# Different Patterns of Care and Survival Outcomes in Transplant-Centre Managed Patients with Early-Stage HCC: Real-World Data from an Australian Multi-Centre Cohort Study

**DOI:** 10.3390/cancers16111966

**Published:** 2024-05-22

**Authors:** Jonathan Abdelmalak, Simone I. Strasser, Natalie L. Ngu, Claude Dennis, Marie Sinclair, Avik Majumdar, Kate Collins, Katherine Bateman, Anouk Dev, Joshua H. Abasszade, Zina Valaydon, Daniel Saitta, Kathryn Gazelakis, Susan Byers, Jacinta Holmes, Alexander J. Thompson, Dhivya Pandiaraja, Steven Bollipo, Suresh Sharma, Merlyn Joseph, Rohit Sawhney, Amanda Nicoll, Nicholas Batt, Myo J. Tang, Stephen Riordan, Nicholas Hannah, James Haridy, Siddharth Sood, Eileen Lam, Elysia Greenhill, John Lubel, William Kemp, Ammar Majeed, John Zalcberg, Stuart K. Roberts

**Affiliations:** 1Department of Gastroenterology, Alfred Health, Melbourne, VIC 3004, Australia; j.abdelmalak@alfred.org.au (J.A.); tangmyojin@gmail.com (M.J.T.); j.lubel@alfred.org.au (J.L.); w.kemp@alfred.org.au (W.K.); a.majeed@alfred.org.au (A.M.); 2Department of Medicine, Central Clinical School, Monash University, Melbourne, VIC 3004, Australia; eileen.lam@monash.edu (E.L.); elysia.greenhill@monash.edu (E.G.); 3Department of Gastroenterology, Austin Hospital, Heidelberg, VIC 3084, Australia; marie.sinclair@austin.org.au (M.S.); avik.majumdar@austin.org.au (A.M.); kate.collins3@austin.org.au (K.C.); kat.bateman@austin.org.au (K.B.); 4AW Morrow Gastroenterology and Liver Centre, Royal Prince Alfred Hospital, Camperdown, NSW 2050, Australia; simone.strasser@health.nsw.gov.au (S.I.S.); natalielyngu@gmail.com (N.L.N.); claude.dennis@health.nsw.gov.au (C.D.); 5Department of Gastroenterology, Monash Health, Clayton, VIC 3168, Australia; anouk.dev@monash.edu (A.D.); joshua.abasszade@monashhealth.org (J.H.A.); 6Department of Gastroenterology, Western Health, Footscray, VIC 3011, Australia; zina.valaydon@wh.org.au (Z.V.); daniel.saitta@wh.org.au (D.S.); kathryn.gazelakis@wh.org.au (K.G.); susan.byers1@wh.org.au (S.B.); 7Department of Gastroenterology, St Vincent’s Hospital Melbourne, Fitzroy, VIC 3065, Australia; jacinta.holmes@svha.org.au (J.H.); alexander.thompson@svha.org.au (A.J.T.); dhivya.pandiaraja@svha.org.au (D.P.); 8Department of Medicine, St. Vincent’s Hospital, University of Melbourne, Parkville, VIC 3052, Australia; 9Department of Gastroenterology, John Hunter Hospital, New Lambton Heights, NSW 2305, Australia; steven.bollipo@health.nsw.gov.au (S.B.); suresh.sharma@health.nsw.gov.au (S.S.); merlyn.joseph@health.nsw.gov.au (M.J.); 10Department of Gastroenterology, Eastern Health, Box Hill, VIC 3128, Australia; rohit.sawhney@easternhealth.org.au (R.S.); amanda.nicoll@easternhealth.org.au (A.N.); nicholas.batt2@austin.org.au (N.B.); 11Department of Medicine, Eastern Health Clinical School, Box Hill, VIC 3128, Australia; 12Department of Gastroenterology, Prince of Wales Hospital, Randwick, NSW 2031, Australia; stephen.riordan@health.nsw.gov.au; 13Department of Gastroenterology, Royal Melbourne Hospital, Parkville, VIC 3052, Australia; nicholas.hannah2@mh.org.au (N.H.); james.haridy@mh.org.au (J.H.); sood.s@unimelb.edu.au (S.S.); 14School of Public Health and Preventative Medicine, Monash University, Melbourne, VIC 3004, Australia; john.zalcberg@monash.edu; 15Department of Medical Oncology, Alfred Health, Melbourne, VIC 3004, Australia

**Keywords:** hepatocellular carcinoma, early, transplant centre, patterns of care, survival

## Abstract

**Simple Summary:**

Little is known about the differences in liver cancer care between liver cancer referral centres, with and without an integrated liver transplant program. Even after adjusting for other important factors, such as tumour stage and patient characteristics, we found that liver transplant centres systematically used different local cancer treatments compared to non-transplant centres. Patients managed at liver transplant centres showed improved overall survival, which was evident even after considering the increased rates of transplantation.

**Abstract:**

The management of early-stage hepatocellular carcinoma (HCC) is complex, with multiple treatment strategies available. There is a paucity of literature regarding variations in the patterns of care and outcomes between transplant and non-transplant centres. We conducted this real-world multi-centre cohort study in two liver cancer referral centres with an integrated liver transplant program and an additional eight non-transplant HCC referral centres across Australia to identify variation in patterns of care and key survival outcomes. Patients with stage Barcelona Clinic Liver Cancer (BCLC) 0/A HCC, first diagnosed between 1 January 2016 and 31 December 2020, who were managed at a participating site, were included in the study. Patients were excluded if they had a history of prior HCC or if they received upfront liver transplantation. A total of 887 patients were included in the study, with 433 patients managed at a liver cancer centre with a transplant program (LTC) and 454 patients managed at a non-transplant centre (NTC). Management at an LTC did not significantly predict allocation to resection (adjusted OR 0.75, 95% CI 0.50 to 1.11, *p* = 0.148). However, in those not receiving resection, LTC and NTC patients were systematically managed differently, with LTC patients five times less likely to receive upfront ablation than NTC patients (adjusted OR 0.19, 95% CI 0.13 to 0.28, *p* < 0.001), even after adjusting for tumour burden, as well as for age, gender, liver disease aetiology, liver disease severity, and medical comorbidities. LTCs exhibited significantly higher proportions of patients undergoing TACE for every tumour burden category, including those with a single tumour measuring 2 cm or less (*p* < 0.001). Using multivariable Cox proportional hazards analysis, management at a transplant centre was associated with reduced all-cause mortality (adjusted HR 0.71, 95% CI 0.51 to 0.98, *p* = 0.036), and competing-risk regression analysis, considering liver transplant as a competing event, demonstrated a similar reduction in risk (adjusted HR 0.70, 95% CI 0.50 to 0.99, *p* = 0.041), suggesting that the reduced risk of death is not fully explained by higher rates of transplantation. Our study highlights systematic differences in HCC care between large volume liver transplant centres and other sites, which has not previously been well-described. Further work is needed to better define the reasons for differences in treatment allocation and to aim to minimise unwarranted treatment variation to maximise patient outcomes across Australia.

## 1. Introduction

Hepatocellular carcinoma (HCC) is the third most common global cause of cancer-related deaths [1], with an estimated 830,000 deaths in 2020 [2]. Continuing to increase in incidence in most parts of the world [3,4], HCC is projected to account for 60% of all deaths due to chronic liver disease by 2030 [5]. HCC is staged using the BCLC algorithm [6], in which tumour burden, liver function, and cancer-related performance status are considered together. The goal of HCC surveillance is to identify HCC in the very early (BCLC 0) and early (BCLC A) stage, when HCC curative treatment modalities such as resection, ablation, and transplantation can be delivered [7]. However, multiple studies have shown poor adherence to standardised management guidelines, with up to 40% of patients with BCLC 0/A stage disease failing to receive upfront curative therapy [8,9,10,11]. Even in those who do receive upfront curative therapy, real-world outcomes leave much to be desired, with 5-year recurrence rates described as between 30–60%, depending on treatment choice, liver disease severity, and tumour burden [12,13,14].

Optimising care of BCLC 0/A stage HCC is therefore a clear priority to maximise patient outcomes. Little is known regarding variation in care between liver transplant centres (LTCs) and non-transplant centres (NTCs). In Australia, LTCs manage a greater volume of HCC patients than do NTCs. Additionally, at LTCs, the multidisciplinary meeting (MDM) involves transplant physicians and surgeons, unlike the MDMs at NTCs. Physicians, surgeons, and interventional radiologists working at an LTC are also exposed to a larger number of patients with more advanced liver disease. We therefore hypothesised that aside from increased rates of transplantation, there may be other differences in patterns of care between LTCs and NTCs, even when considering only BCLC 0/A patients and adjusting for clinical covariates. We performed this real-world multi-centre retrospective study to test this hypothesis and additionally, to investigate whether there was any variation in survival outcomes between the two cohorts after adjusting for key clinical variables.

## 2. Materials and Methods

### 2.1. Participants

Our study involved participants with BCLC stage 0/A HCC, with a first diagnosis of HCC between 1 January 2016 and 31 December 2020, at two Australian LTCs and an additional eight NTCs across Victoria and New South Wales. Patients were eligible for the study if they met the following inclusion criteria: adult aged > 18 years of age; diagnosis of HCC documented between 1 January 2016 and 31 December 2020 on the basis of imaging fulfilling LI-RADS 5 criteria or histology confirming HCC; and confirmed BCLC 0 or A disease, with complete documentation of single lesion of any size or up to three lesions, with no lesions > 3 cm, Child–Pugh (CP) class A or B, an ECOG cancer-related performance status of 0, and the absence of extrahepatic disease or macrovascular invasion. Exclusion criteria were: receipt of upfront liver transplantation; prior diagnosis or past history of HCC; diagnosis of other solid organ malignancy, other than non-melanotic skin cancer; and insufficient data to determine stage of HCC, treatment, or follow-up.

Waiver of consent was sought, with all patient data entered in a deidentified form. Ethics for the study were approved by Monash University and the Human Research Ethics Committee (HREC), located at each respective site.

### 2.2. Study Design

This was a multi-centre retrospective cohort study. Data was collected retrospectively from the patient’s medical records, from the date of initial diagnosis of HCC to the end of follow-up (either death or last medical record entry available at time of data extraction). The minimum dataset is outlined in full in Appendix B (Table A1). Data included key clinical variables, tumour characteristics, and biochemistry at time of diagnosis, as well as initial treatment and subsequent assessments of treatment response, and sequential treatments with subsequent treatment response. Modified RECIST criteria (mRECIST) were used at all sites to describe treatment response after initial treatment and at subsequent follow-up. Treatments were characterised as resection, ablation (including microwave ablation, radiofrequency ablation, and percutaneous ethanol injection), transarterial chemoembolisation (TACE, including the conventional treatment or with drug-eluting beads), or other (bland hepatic artery embolization, selective internal radiation therapy, stereotactic body radiation therapy, systemic therapy). For survival analysis, the date of diagnosis was considered as the index date. All data was de-identified and entered into a centralised database, using a REDCap electronic data capture system hosted at Monash University.

### 2.3. Statistical Analysis

Subjects were considered as an ‘LTC’ or ‘NTC’ patient, depending on their initial managing centre. NTC patients who went on to be referred to LTC and underwent subsequent treatment or transplantation during follow-up were still considered as NTC patients for the purposes of our study. Data were analysed using SPSS 29.0 (SPSS, Inc., Chicago, IL, USA) and STATA 18 (StataCorp. 2023, College Station, TX, USA) software. Categorical variables were described using a frequency table, and a Chi-square test was used to test statistical significance between the two groups. Non-parametric continuous variables were summarised using the median and interquartile range, and Mann−Whitney U test was used as the test of statistical significance. Parametric continuous variables were reported with mean and standard deviation, and the independent samples *t*-test was used to test for statistical significance in comparing the two groups. Binary logistic regression was used to assess the factors predicting treatment allocation to resection and to ablation. We did not perform such analysis on treatment allocation to transplantation during follow-up; as multiple variables predicting transplant suitability were not captured (in particular, recency of alcohol intake, patients’ wishes, presence of social support, and medical/psychological comorbidities not captured in the CCI), reverse causality could not be excluded as the leading cause for the difference in treatment allocation, and there were overall small numbers of transplanted patients, limiting statistical power. Multivariable Cox proportional hazards analysis was used to assess for predictors of all-cause death, with calculation of adjusted hazard ratios with 95% confidence intervals estimated for each inputted clinical variable and cumulative hazard function curves plotted. Variables were selected based on their clinical significance, with all variables included in the final model, including those that were statistically insignificant. To reduce the risk of competing-risk bias due to liver transplantation, which is known to significantly reduce risk of death, we performed competing-risks regression analysis, with liver transplantation as a competing risk for all-cause death. Adjusted hazard ratios, with 95% confidence intervals for each of the variables, as well as cumulative incidence function curves, were produced using the STATA 18 statistical software. In all tests of statistical significance performed, two-tailed *p* < 0.05 was deemed a statistically significant difference.

## 3. Results

### 3.1. Patients

Across all sites, 887 patients were eligible for inclusion, with 433 patients from the two LTCs and 454 patients from the remaining eight NTCs. Patient characteristics, in the distinct LTC and NTC populations, are presented in Table 1. Both LTC and NTC patients exhibited a similar male preponderance. LTC patients were significantly younger than NTC patients (mean age 63.6 vs. 65.5, *p* = 0.011). There were slight differences in liver disease aetiology, with NTC patients more likely to have HBV and alcohol as a cause of their liver disease. Medical comorbidities, measured using the Charlson Comorbidity Index (CCI), were similar between the two cohorts. Median platelet count was significantly lower in the LTC cohort compared to the NTC cohort (117 vs. 137, *p* < 0.001) suggesting greater prevalence and severity of portal hypertension on average in the LTC group. Similarly, LTC patients were more likely to exhibit more severe liver disease, with a greater proportion of Child−Pugh (CP) B patients (*p* = 0.033). Tumour burden was significantly different (*p* = 0.024) between the two groups, with LTC patients more likely to have both multinodular disease and large single tumours and NTC patients more likely to have small solitary tumours. A greater proportion of NTC patients underwent resection as the initial treatment, likely reflecting the higher numbers of patients without clinically significant portal hypertension. A significantly greater subset of LTC patients underwent initial TACE compared to those from the NTC (256 vs. 111, *p* < 0.001), but patients who did receive initial TACE were more likely to undergo ablation as a second follow-up treatment at LTCs compared to at NTCs (41.4% vs. 25.2%, *p* = 0.016), suggesting a differing treatment strategy. No patients who received initial TACE went on to receive resection later.

### 3.2. Transplantation

Of the 887 patients included in our study, a total of 42 patients (4.7%) underwent transplantation during follow-up. Seven patients who were initially managed at an NTC were referred to a transplant centre and underwent transplantation during the period of observation in our study. The characteristics of patients who underwent transplant during follow-up are outlined in Table 2. The vast majority of patients who received transplantation did so after a recurrence after a documented complete response (CR), rather than as a salvage curative treatment (38 vs. 4). Patients originally from an NTC who underwent transplantation during follow-up were more likely to have a lower initial CCI on average, as well as to receive a non-TACE initial treatment. Otherwise, NTC and LTC patients who underwent transplantation were similar with respect to age, sex, liver disease aetiology, CPS, and initial tumour burden. The time from initial diagnosis to receipt of transplant was similar between the two groups (median 504 days vs. 729 days, *p* = 0.446).

### 3.3. Resection

Of the 887 patients included in our study, a total of 199 patients (22.4%) underwent surgical resection as the initial treatment. The patient characteristics of those who received resection in the LTC and NTC cohorts are outlined in Table 3. There was no evidence of a systematic difference in any clinical characteristics, suggesting similar patient selection across all sites. In particular, LTC and NTC patients who underwent resection had similar platelet counts (median 161 vs. 194, *p* = 0.068), similar CCI (median 3 vs. 3, *p* = 0.372), and nearly all had Child–Pugh A5 or A6 liver disease (96.2% vs. 97.5%, *p* = 0.893).

Table 4 presents a multivariable binary logistic regression assessing treatment allocation to resection vs. non-resection treatment in all 887 patients. Importantly, management at an LTC treatment site had no significant association with treatment allocation to resection (adjusted OR 0.748, 95% CI 0.502 to 1.114, *p* = 0.153). As expected, the presence of cirrhosis predicted against treatment allocation to resection (adjusted OR 0.346, 95% CI 0.210 to 0.570, *p* < 0.001). Correspondingly, higher platelet counts predicted treatment allocation to resection (adjusted OR 1.004, 95% CI 1.001 to 1.007, *p* = 0.003). Higher CPS and CCI both predicted against treatment allocation to resection (adjusted OR 0.504, 95% CI 0.370 to 0.685, *p* < 0.001; and adjusted OR 0.728, 95% CI 0.638 to 0.831, *p* < 0.001 respectively).

### 3.4. Locoregional Treatment

Of the remaining 689 patients in our study who did not receive surgical resection as the initial treatment, we found significant differences in treatment strategy between LTCs and NTCs. Table 5 presents the initial treatment choice between LTCs and NTCs for varying tumour burdens. Significant differences in treatment allocation were seen in all tumour burden categories. In patients with a single tumour measuring 2 cm or less, the majority of patients at the NTC underwent upfront ablation (77%), whereas most LTC patients underwent TACE as their first treatment (60%). Similarly, in those with a single tumour measuring 3 cm or less and larger than 2 cm, NTC patients mainly received ablation (56%), while LTC patients mainly received TACE (63%). For those with a single tumour larger than 3 cm, no LTC patients received upfront ablation, with most receiving TACE, whereas NTC continued to offer attempts at upfront ablation in small numbers in these patients. In those with a multinodular tumour within the BCLC A criteria (all nodules 3 cm or less), NTC offered upfront ablation to a sizeable minority of patients (46%), whereas LTC only proceeded to ablation first in 10% of patients.

Multivariable binary logistic regression, assessing only the population who did not receive resection (n = 689), was performed to examine the predictors of initial treatment allocation to ablation vs. non-ablation. The results are presented in Table 6. After adjusting for key covariates, management at LTC was associated with approximately a five-fold reduced chance of undergoing ablation as the initial treatment (adjusted OR 0.19, 95% CI 0.13 to 0.28, *p* < 0.001). Increased CPS was also associated with a reduced chance of allocation to ablation (adjusted OR 0.82, 95% CI 0.69 to 0.99, *p* = 0.037). As expected, compared to the presence of a single tumour 2 cm or less, larger tumour sizes or the presence of multinodular tumours were associated with a reduced chance of allocation to ablation. Age, sex, liver disease aetiology, CCI, and platelet count did not predict for or against treatment allocation to ablation.

### 3.5. All-Cause Mortality

Median follow-up time from initial diagnosis to death or censorship was 3.39 years in the LTC group and 3.83 years in the NTC group (*p* = 0.31). a multivariable Cox proportional hazards analysis was performed on the entire cohort, and multivariable-adjusted hazard function curves are presented in Figure 1. Management at LTCs was associated with a reduced risk of mortality over the course of follow-up (adjusted HR 0.71, 95% CI 0.51 to 0.98, *p* = 0.036) despite adjusting for age, sex, smoking, diabetes, HBV, alcohol, cirrhosis status, CCI, platelet count, CPS and tumour burden category. Female sex was associated with reduced risk of death (adjusted HR 0.63, 95% CI 0.40 to 0.99, *p* = 0.044), while an increased risk of death was associated with CCI (adjusted HR 1.17, 95% CI 1.08 to 1.28, *p* < 0.001), CPS (adjusted HR 1.44, 95% CI 1.26 to 1.64, *p* < 0.001) and single tumour >5 cm (adjusted HR 2.22, 95% CI 1.20 to 4.09, *p* = 0.011). Age, smoking, diabetes, HBV, alcohol, cirrhosis status, platelet count, and multinodular tumour did not significantly affect all-cause mortality in the multivariable adjusted model. The model is presented in detail in Appendix A.

### 3.6. All-Cause Mortality with Liver Transplant as Competing Risk

Competing-risks regression analysis was performed on the entire cohort. Multivariable-adjusted incidence function curves are presented in Figure 2. After including liver transplant as a competing event, there was evidence of a similarly significant reduction in risk in LTC patients (adjusted HR 0.71., 95% CI 0.50 to 0.99, *p* = 0.041) with adjustment for age, sex, smoking, diabetes, HBV, alcohol, cirrhosis status, CCI, platelet count, CPS, and tumour burden category. Female sex was associated with reduced risk of transplant or death (adjusted HR 0.63, 95% CI 0.40 to 0.98, *p* = 0.042). Predictors of transplant or death included CCI (adjusted HR 1.17, 95% CI 1.07 to 1.29, *p* = 0.001), CPS (adjusted HR 1.43, 95% CI 1.25 to 1.64, *p* < 0.001) and a single tumour >5 cm (adjusted HR 2.17, 95% CI 1.17 to 4.01, *p* = 0.014). The results of the competing-risks regression analysis are presented in detail in Appendix A. Notably, other tumour burden categories, age, smoking, diabetes, HBV, alcohol, cirrhosis status, platelet count were non-significant predictors.

## 4. Discussion

HCC management is complex and highly individualised, with treatment decisions made based not just on BCLC stage alone but also using specific anatomical considerations, non-liver medical comorbidities, patient views and values, and centre-specific experience and resources. Variation in care across centres has been well described [15,16], but there is no published data, to our knowledge, regarding specifically differences between LTCs and NTCs and no published work describing variations in Australia. Our study is the first to show, in an Australian context, that there are systematic differences in the treatment approach between LTCs and NTCs, even after adjusting for the differences in patient characteristics between the two cohorts. Furthermore, our study demonstrated that patients receiving treatment for BCLC 0/A HCC at an LTC exhibit reduced all-cause mortality, primarily driven by the survival benefit associated with transplantation.

As expected, we found significant and systematic differences in the population between those managed at LTCs and NTCs. Patients managed at LTCs in Australia are comprised of a combination of those for whom the LTC is their closest hospital and others who were referred from further away for tertiary management of liver disease or HCC. We expect that those referred for tertiary management to a LTC rather than NTC are the reasons for the systematic differences, with referrers more likely to refer transplant-suitable patients to LTCs, even when there is no indication for transplant at the time of initial referral. The first of the observed differences was in liver disease aetiology, with LTC patients less likely to exhibit alcohol as a cause for their underlying liver disease. This may be due to patients with ongoing active drinking being more likely to be referred for HCC treatment at an NTC rather than an LTC. LTC patients were also less likely to show HBV as a cause for their liver disease. Firstly, we suspect that this is due to non-cirrhotic CHB patients being managed more frequently at an NTC, for both unique Australian geographic reasons, in which many of the NTCs in our study serve communities with higher proportions of Southeast Asian migrants, as well as for clinical reasons, where low-risk resection patients are less likely to be referred to an LTC. Secondly, we found that the LTCs possessed a younger patient cohort compared to the NTCs, with a greater variation in age seen in the NTC population. This is due to a greater number of elderly patients observed in the NTC population, where these patients are also less likely to be referred to LTC due to their advanced age. Thirdly, liver disease was clearly more severe in the LTC cohort, with the lower median platelet count reflecting more significant portal hypertension and higher Child–Pugh scores, indicating more diminished hepatic reserve. We suspect that this reflects the diversion of patients with more advanced liver disease to the LTC from the NTC, as well as the higher proportion of non-cirrhotic CHB patients in the NTC. Lastly, LTC patients exhibited higher risk tumour burden categories, with greater numbers of large single tumours or multinodular tumours. Again, we expect that this is due to these patients being more likely to be referred to an LTC for management.

While a smaller proportion of LTC compared to NTC patients underwent resection (18.2% vs. 26.4%), after adjusting for the clinical predictors of treatment allocation to resection, patients were just as likely to undergo resection at LTCs as those managed at NTCs. There was a numerical signal that patients undergoing resection at an LTC had a lower median platelet count than did NTC patients, suggesting that LTCs may be offering resection to higher-risk patients, but this result was not statistically significant (median platelet 161 vs. 194, *p* = 0.068). There were otherwise no significant differences between the two groups, suggesting similar patient selection for resection irrespective of management at an LTC or an NTC. As resection is the most efficacious curative first line treatment, aside from transplantation, and access to resection has been defined as an HCC quality indicator [17,18], our results reassure us that access to resection is not centre-dependent in our study population.

After adjusting for all key clinical variables, management at an LTC was independently associated with a 5-fold reduced chance of upfront ablation. We found that in patients not deemed suitable for resection, LTCs instead favoured initial TACE across all tumour burden categories. Almost one-third of LTC patients who received TACE as their initial therapy at an LTC underwent ablation as a second therapy, with an additional 10% undergoing ablation as a third or subsequent treatment. The BCLC treatment algorithm recommends upfront ablation as the locoregional curative treatment choice for BCLC 0/A patients [19], and none of the major international or Australian guidelines recommend TACE as initial therapy in BCLC 0 or A disease [6,20,21,22]. However, a growing body of literature published over the last six years has described the beneficial effect of TACE prior to ablation [23,24,25], with improved outcomes postulated to be due to elimination of viable micrometastases adjacent to the tumour, as well as disruption of hepatic arterial flow, leading to a reduced heat sink effect at the time of ablation, increasing the size of the ablation zone, making it more likely for the treatment to be complete and efficacious. Furthermore, the intra-tumoural deposition of lipiodol with TACE can improve tumour visualisation at the time of ablation, increasing the likelihood of optimal ablation needle placement [23,24,25]. Further work is needed to determine whether a treatment approach with upfront ablation or initial TACE followed by ablation significantly affects outcomes, particularly if there is a delay between initial TACE and subsequent ablation therapy, as was the case for the majority of patients in our study. Beyond the benefits of TACE prior to ablation, the treatment choice of initial TACE vs. upfront ablation may also be biased by transplant eligibility (such as more aggressive pursuit of cure in those not eligible for transplantation or more judicious treatment with an emphasis on safety in those considered transplant candidates), although this is purely speculative and requires prospective evaluation.

In our study we did find that across all patients, there was superior overall survival in the LTC cohort. Patients managed at LTCs were 29% less likely to die over the course of follow-up, after adjusting for clinical factors at presentation. Even after considering liver transplant as a competing risk, with the same multivariable adjustment, LTC patients had a similar 30% reduced risk of death. The reason for the reduced risk of death is not certain with the difference in treatment strategy observed between LTCs and NTCs in non-resection treatments potentially being a key contributing factor. Notably, confounding from invisible patient factors as previously discussed above are another explanation, as these are likely systematically different between LTCs and NTCs and affect not just access to transplantation but also access to other cancer treatments as well as influence the risk of hepatic decompensation. Furthermore, survival analysis may be affected by guarantee-time bias, which we have been unable to mitigate due to the lack of granularity in the data regarding movement of patients between managing centres. Prospective assessment of similar patients managed at LTCs and NTCs is needed to further investigate our findings.

In comparing the patients who received transplant during follow-up, patients at LTCs and NTCs exhibited overall similar characteristics, suggesting similar patient selection. Specifically, the two groups were similar with respect to age, sex, liver disease aetiology, Child–Pugh Score, and tumour burden category. The one area in which the two groups differed was in CCI, where LTC patients had a slightly higher CCI, which was statistically significant, raising the possibility that patients may be more likely to proceed to transplantation, with greater medical comorbidities, if their initial managing centre is an LTC. Importantly, the time to transplant from initial diagnosis was not significantly different between the two groups, suggesting that where LT was required, patients being referred from an NTC were not subjected to significant delays. It is important to note that as a retrospective study, reverse causality is an important consideration for the larger numbers of LTC patients who received transplantation. Important factors such as recency of alcohol intake, social situation, patient wishes, and other medical/psychological comorbidities not captured by the CCI may all have contributed to an individual patient’s suitability for transplant, and these factors have not been captured in our data collection and therefore remain unadjusted for. It should also be noted that while some patients have been considered as LTC patients in our study, they may have actually received their initial diagnosis in the community, in the private health care sector, or at a small peripheral hospital and may have been selected for LTC over NTC referral due to their perceived transplant suitability, despite not requiring transplant at the time of initial referral. Indeed, between 65% and 75% of patients were referred to centres from outside their direct local hospital catchment, suggesting that selection bias in the preferential referral of transplant-suitable patients to an LTC in preference to an NTC played a significant role in producing distinct populations, with resultant differences in transplant suitability. Explicit data regarding referral origin, time at managing centre prior to HCC diagnosis, clear documentation regarding perceived transplant suitability, and transplant waitlisting were not available for analysis in our study. We are therefore unable to distinguish between LTC patients who were referred specifically for transplant and those who were referred solely for HCC management as the most-suitable HCC referral centre. Overall, we do not believe that our study provides reliable evidence to form conclusions regarding differential access to transplantation, but instead, it provides some early insights that can be further investigated prospectively. Equitable access to transplantation is a major goal in maximizing the quality of HCC care across the nation, and we will report our prospective results in due course.

Our study has several strengths. Firstly, we report real-world data giving insights into the day-to-day patterns of care and outcomes of Australian early-stage HCC patients across multiple centres. Secondly, we were able to adjust for the majority of important factors in considering both treatment allocation and survival outcomes, increasing confidence regarding the authenticity of the differences observed between LTCs and NTCs. Lastly, we have reported on a large population, allowing us the statistical power required to observe the significant differences in locoregional treatment strategy and overall survival.

As alluded to above, our study also has significant limitations. Firstly, because our study is retrospective in nature, it is susceptible to information bias, particularly in association with clinical factors and treatment allocation. Secondly, differences in the survival outcomes observed may be affected by selection bias, both direct and indirect, by confounding and in our competing-risks regression, by guarantee-time bias. The use of a multivariable model to mitigate this increases our confidence in the findings, but some factors, such as the anatomical location of the tumour or ongoing alcohol intake, have not been captured. Future studies would benefit from collecting explicit information regarding eligibility for liver transplantation to better evaluate reasons for differences in treatment selection between LTC and NTC.

## 5. Conclusions

Our study provides valuable evidence that there are indeed systematic differences in patterns of care in BCLC 0/A HCC between HCC referral centres, with and without an integrated liver transplant program, with the main difference involving difference in preference between upfront ablation and initial TACE in patients not suitable for resection. While patients managed at LTCs have improved overall survival, even after considering transplant as a competing-risk, it is not certain if the difference in treatment strategy is the direct cause for this result. Further work is needed to prospectively evaluate the differences in treatment strategy, access to transplantation, and long-term survival outcomes, both within and across centres, in order to identify opportunities for quality improvement.

## Figures and Tables

**Figure 1 cancers-16-01966-f001:**
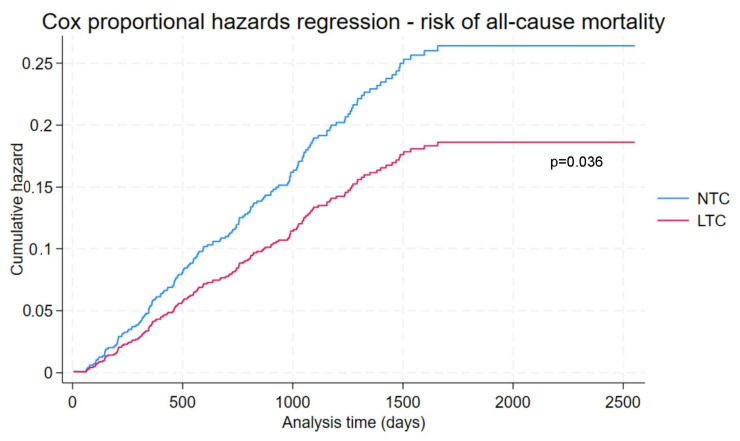
All-cause mortality hazard function after multivariable adjustment.

**Figure 2 cancers-16-01966-f002:**
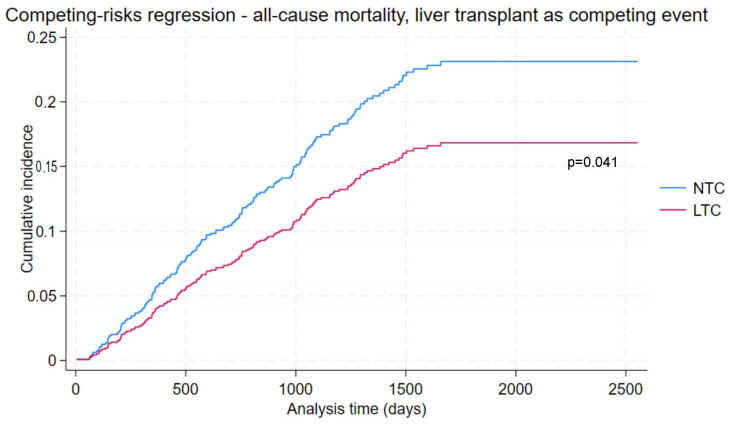
Competing-risks regression with cumulative incidence function of all-cause mortality, with liver transplant as a competing risk.

**Table 1 cancers-16-01966-t001:** Patient characteristics across liver transplant and non-transplant centres.

	LTC *n* = 433	NTC *n* = 454	*p*-Value
Age *	63.6 ± 10.0	65.5 ± 11.7	0.011
Sex Male Female	351 (81.1%) 82 (18.9%)	362 (79.7%) 92 (20.3%)	0.247
Aetiology Alcohol HBV HCV MASLD Other metALD HBV/HCV HCV + SLD HBV + SLD	62 (14.3%) 53 (12.2%) 74 (17.1%) 49 (11.3%) 28 (6.5%) 26 (6.0%) 13 (3.0%) 119 (27.5%) 9 (2.1%)	72 (15.9%) 57 (12.6%) 66 (14.5%) 63 (13.9%) 26 (5.7%) 29 (6.4%) 19 (4.2%) 95 (20.9%) 27 (5.9%)	0.047
Smoking Yes No	102 (76.4%) 331 (23.6%)	151 (66.7%) 303 (33.3%)	0.001
Charlson Comorbidity Index **	5 (3 to 6)	4 (3 to 6)	0.155
Cirrhosis Yes No	369 (85.2%) 64 (14.8%)	372 (81.9%) 82 (18.1%)	0.188
Platelets **	117 (80 to 164)	137 (92 to 204)	<0.001
Child–Pugh Score 5 6 7 8 9	243 (56.1%) 100 (23.1%) 44 (10.2%) 25 (5.8%) 21 (4.8%)	248 (54.6%) 132 (29.1%) 48 (10.6%) 17 (3.7%) 9 (2.0%)	0.033
Tumour Burden Category Single ≤ 2 cm Single > 2 cm, ≤3 cm Single > 3 cm, ≤5 cm Single > 5 cm Multinodular, all ≤ 3 cm	121 (27.9%) 95 (21.9%) 69 (15.9%) 46 (10.6%) 102 (23.6%)	141 (31.1%) 129 (28.4%) 72 (15.9%) 35 (7.7%) 77 (17.0%)	0.024
Initial Treatment Allocation Resection Ablation TACE Other	79 (18.2%) 74 (17.1%) 256 (59.1%) 24 (5.5%)	120 (26.4%) 185 (40.7%) 111 (24.4%) 38 (8.4%)	<0.001
Follow-up ablation after TACE After first TACE After second TACE After third or subsequent TACE No ablation during follow-up	80 (31.3%) 12 (4.7%) 14 (5.5%) 150 (58.6%)	18 (16.2%) 6 (5.4%) 4 (3.6%) 83 (74.8%)	0.016

* mean ± standard deviation; ** median (25th percentile to 75th percentile). LTC, liver transplant centre; NTC, non-transplant centre; HBV, hepatitis B virus; HCV, hepatitis C virus; MASLD, metabolic dysfunction-associated steatotic liver disease; metALD, metabolic and alcohol related liver disease; SLD, steatotic liver disease; TACE, transarterial chemoembolisation.

**Table 2 cancers-16-01966-t002:** Patient characteristics of those who underwent liver transplantation during follow-up.

	LTC n = 35	NTC n = 7	*p*-Value
Age *	59.8 ± 5.1	53.7 ± 7.0	0.062
Sex Male Female	32 (91.4%) 3 (8.6%)	5 (71.4%) 2 (28.6%)	0.136
Aetiology Alcohol HBV HCV MASLD Other metALD HBV/HCV HCV + SLD	4 (11.4%) 4 (11.4%) 3 (8.6%) 4 (11.4%) 1 (2.9%) 4 (11.4%) 1 (2.9%) 14 (40.0%)	0 3 (42.9%) 1 (14.3%) 0 1 (14.3%) 0 1 (14.3%) 1 (14.3%)	0.170
Charlson Comorbidity Index **	5 (4 to 6)	3 (2 to 4)	0.028
Initial Child–Pugh Score 5 6 7 8 9	16 (45.7%) 5 (14.3%) 3 (8.6%) 4 (11.4%) 7 (20.0%)	3 (42.9%) 2 (28.6%) 2 (28.6%) 0 0	0.299
Initial Tumour Burden Category Single ≤ 2 cm Single > 2 cm, ≤3 cm Single > 3 cm, ≤5 cm Single > 5 cm Multinodular, all ≤ 3 cm	8 (22.9%) 9 (25.7%) 4 (11.4%) 2 (5.7%) 12 (34.3%)	1 (14.3%) 2 (28.6%) 0 0 4 (57.1%)	0.696
Initial Treatment Resection Ablation TACE Other	3 (8.6%) 5 (14.3%) 27 (77.1%) 0	1 (14.3%) 4 (57.1%) 1 (14.3%) 1 (14.3%)	0.003
Indication for Transplant Salvage Recurrence	3 (8.6%) 32 (91.4%)	1 (14.3%) 6 (85.7%)	0.562
Time from initial diagnosis to transplant (days) **	504 (349 to 1019)	729 (546 to 743)	0.446

* mean ± standard deviation; ** median (25th percentile to 75th percentile). LTC, liver transplant centre; NTC, non-transplant centre; HBV, hepatitis B virus; HCV, hepatitis C virus; MASLD, metabolic dysfunction-associated steatotic liver disease; metALD, metabolic and alcohol related liver disease; SLD, steatotic liver disease; TACE, transarterial chemoembolization.

**Table 3 cancers-16-01966-t003:** Patient characteristics of those who received surgical resection.

	LTC n = 79	NTC n = 120	*p*-Value
Age *	64.0 ± 8.5	62.7 ± 10.3	0.346
Sex Male Female	62 (78.5%) 17 (21.5%)	95 (79.2%) 25 (20.8%)	0.908
Aetiology Alcohol HBV HCV MASLD Other metALD HBV/HCV HCV + SLD HBV + SLD	10 (12.7%) 21 (26.6%) 18 (22.8%) 8 (10.1%) 5 (6.3%) 3 (3.8%) 2 (2.5%) 12 (15.2%) 0	8 (6.7%) 30 (25.0%) 18 (15.0%) 15 (12.5%) 8 (6.7%) 3 (2.5%) 3 (2.5%) 25 20.8(%) 10 (8.3%)	0.178
Smoking Yes No	18 (22.8%) 61 (77.2%)	41 (34.2%) 79 (65.8%)	0.085
Charlson Comorbidity Index **	3 (2 to 4)	3 (2 to 4)	0.372
Cirrhosis Yes No	43 (54.4%) 36 (45.6%)	72 (60.0%) 48 (40.0%)	0.436
Platelets **	161 (131 to 216)	194 (142 to 242.5)	0.068
Child–Pugh Score 5 6 7 8	66 (83.5%) 10 (12.7%) 2 (2.5%) 1 (1.3%)	98 (81.7%) 19 (15.8%) 2 (1.7%) 1 (0.8%)	0.893
Tumour Burden Category Single ≤ 2 cm Single > 2 cm, ≤3 cm Single > 3 cm, ≤5 cm Single > 5 cm Multinodular, all ≤ 3 cm	19 (24.1%) 22 (27.8%) 17 (21.5%) 16 (20.3%) 5 (6.3%)	32 (26.7%) 35 (29.2%) 34 (29.3%) 12 (28.3%) 7 (6.8%)	0.331

* mean ± standard deviation; ** median (25th percentile to 75th percentile). LTC, liver transplant centre; NTC, non-transplant centre; HBV, hepatitis B virus; HCV, hepatitis C virus; MASLD, metabolic dysfunction-associated steatotic liver disease; metALD, metabolic and alcohol related liver disease; SLD, steatotic liver disease; TACE, transarterial chemoembolization.

**Table 4 cancers-16-01966-t004:** Multivariable binary logistic regression-predictors of allocation to surgical resection vs. non-surgical treatment.

	Adjusted OR	95% CI	*p*-Value
Centre Type			
NTC	Reference	−	−
LTC	0.75	0.50 to 1.11	0.153
Age	1.00	0.98 to 1.02	0.922
Sex			
Male	Reference	−	−
Female	1.22	0.75 to 1.99	0.432
Diabetes			
No	Reference	−	−
Yes	0.69	0.41 to 1.17	0.170
Smoking			
No	Reference	−	−
Yes	1.30	0.82 to 2.06	0.262
HBV			
No	Reference	−	−
Yes	1.14	0.72 to 1.81	0.571
Alcohol			
No	Reference	−	−
Yes	0.87	0.55 to 1.36	0.535
Charlson Comorbidity Index	0.73	0.64 to 0.83	<0.001
Cirrhosis			
No	Reference	−	−
Yes	0.35	0.21 to 0.57	<0.001
Platelets	1.00	1.00 to 1.01	0.003
Child–Pugh Score	0.50	0.37 to 0.69	<0.001
Tumour Burden Category			
Single ≤ 2 cm	Reference	−	−
Single > 2 cm, ≤3 cm	1.35	0.82 to 2.22	0.233
Single > 3 cm, ≤5 cm	2.03	1.16 to 3.55	0.013
Single > 5 cm	0.94	0.47 to 1.88	0.855
Multinodular, all ≤ 3 cm	0.35	0.17 to 0.73	0.004

LTC, liver transplant centre; NTC, non-transplant centre; HBV, hepatitis B virus.

**Table 5 cancers-16-01966-t005:** Choice of initial treatment by tumour burden category in those not undergoing resection.

	LTC n = 354	NTC n = 334	*p*-Value
Single ≤ 2 cm Ablation TACE Other	41 (40.2%) 61 (59.8%) 0	84 (77.1%) 16 (14.7%) 9 (8.3%)	<0.001
Single > 2 cm, ≤3 cm Ablation TACE Other	23 (31.5%) 46 (63.0%) 4 (5.5%)	53 (56.4%) 24 (25.5%) 17 (18.1%)	<0.001
Single > 3 cm, ≤5 cm Ablation TACE Other	0 49 (94.2%) 3 (5.8%)	13 (34.2%) 23 (60.5%) 2 (52.6%)	<0.001
Single > 5 cm Ablation TACE Other	0 16 (53.3%) 14 (46.7%)	3 (13.0%) 13 (56.5%) 7 (30.4%)	0.091
Multinodular, all ≤ 3 cm Ablation TACE Other	10 (9.7%) 84 (86.6%) 3 (3.1%)	32 (45.7%) 35 (50.0%) 3 (4.3%)	<0.001

LTC, liver transplant centre; NTC, non-transplant centre; TACE, transarterial chemoembolisation.

**Table 6 cancers-16-01966-t006:** Multivariable binary logistic regression-predictors of allocation to upfront ablation vs. other treatment in those not undergoing resection.

	Adjusted OR	95% CI	*p*-Value
Centre Type			
NTC	Reference	−	−
LTC	0.19	0.13 to 0.28	<0.001
Age	1.00	0.98 to 1.02	0.799
Sex			
Male	Reference	−	−
Female	1.07	0.67 to 1.70	0.789
Diabetes			
No	Reference	−	−
Yes	1.05	0.68 to 1.61	0.835
Smoking			
No	Reference	−	−
Yes	1.29	0.84 to 1.96	0.242
HBV			
No	Reference	−	−
Yes	0.97	0.58 to 1.63	0.904
Alcohol			
No	Reference	−	−
Yes	0.96	0.64 to 1.45	0.854
Charlson Comorbidity Index	1.02	0.91 to 1.14	0.740
Cirrhosis			
No	Reference	−	−
Yes	1.94	0.92 to 4.08	0.082
Platelets	1.00	1.00 to 1.00	0.800
Child–Pugh Score	0.82	0.69 to 0.99	0.037
Tumour Burden Category			
Single ≤ 2 cm	Reference	−	−
Single > 2 cm, ≤3 cm	0.50	0.32 to 0.79	0.003
Single > 3 cm, ≤5 cm	0.10	0.05 to 0.21	<0.001
Single > 5 cm	0.04	0.01 to 0.13	<0.001
Multinodular, all ≤ 3 cm	0.22	0.14 to 0.36	<0.001

LTC, liver transplant centre; NTC, non-transplant centre; HBV, hepatitis B virus.

## Data Availability

The data presented in this study are available on request from the corresponding author.

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
