# Peer review of "Different Patterns of Care and Survival Outcomes in Transplant-Centre Managed Patients with Early-Stage HCC: Real-World Data from an Australian Multi-Centre Cohort Study"

_cancers, 2024, doi:10.3390/cancers16111966_

Round 1

Reviewer 1 Report

Comments and Suggestions for Authors

Congratulations to the authors for this study which reports a known cross-section of real life but rarely scientifically demonstrated. This makes the work very interesting even if the results do not reveal anything extraordinary.

I have some observations:

1) The examined sample is well represented and distributed but in my opinion it should be made clearer through a flow chart that explains the various treatments (LT, resection, loco-regional) and any overlaps in the same patient; the division into paragraphs is good to facilitate the reading of the results but the overall vision of the numbers is lost.

2) I didn't understand  well the sentences 188-191 and 198-199, please explain better. Probably, the definition of follow-up is not clear: follow-up after first referral to the centre, follow-up after first treatment, follow-up after transplant? Please, explain better.

3) I didn't understand well the sentences  202-203, please explain better

4) the discussion in my opinion is too long, the conclusions are clear so it is not necessary to comment on all the results because they lead to the same conclusions. Therefore I advise the authors to lighten the discussion, focusing on the main results and those on which interpretation is really necessary

Comments on the Quality of English Language

Nothing to comment, even if probably a minor revision could help the authors to explain some steps better

Author Response

1) The examined sample is well represented and distributed but in my opinion it should be made clearer through a flow chart that explains the various treatments (LT, resection, loco-regional) and any overlaps in the same patient; the division into paragraphs is good to facilitate the reading of the results but the overall vision of the numbers is lost.

Throughout our paper we have focused on initial treatment allocation as all of our data around covariates (tumour number, size, liver disease severity etc) is taken only at time of diagnosis prior to the initial treatment, with no time-varying data over the course of follow up. Because of the variance in response and outcome trajectory after initial treatment (ie. CR with subsequent local recurrence, CR with distant recurrence, PR or SD or PD), we believe it would be misleading to present a flow chart (such as a Sankey diagram) in contrasting LTC and NTC cases, as the difference in treatment response impacting sequential treatment between the two cohorts who would be unexplored and consequently unadjusted for potentially leading to inappropriate comparisons.

Instead, we have summarised the distribution of initial treatment allocation between all LTC and NTC patients in table 1 and summarised the distribution of initial treatment allocation between LTC and NTC patients who underwent transplant during follow up in table 2.

Because TACE is a non-curative modality (unlike ablation and resection), we chose to give an indication regarding subsequent treatment with ablation in Table 1. No patients who underwent initial TACE went on to receive resection – we have made this clear with the addition of the sentence “No patients who received initial TACE went on to receive resection later.” Patients who initially received TACE and went on to transplant are summarised in Table 2.

2) I didn't understand well the sentences 188-191 and 198-199, please explain better. Probably, the definition of follow-up is not clear: follow-up after first referral to the centre, follow-up after first treatment, follow-up after transplant? Please, explain better.

We have changed the phrase “during follow-up” to “as a second follow-up treatment”.

3) I didn't understand well the sentences 202-203, please explain better

We have changed the phrase “over the period of follow up” to during the period of observation in our study”.

4) The discussion in my opinion is too long, the conclusions are clear so it is not necessary to comment on all the results because they lead to the same conclusions. Therefore I advise the authors to lighten the discussion, focusing on the main results and those on which interpretation is really necessary

We believe that all elements included in our discussion are critically relevant to fully understanding the results of our study. Additionally, our discussion contains 1700 words and eight paragraphs which we believe is of reasonable length.

Reviewer 2 Report

Comments and Suggestions for Authors

The authors present a well conceived and executed study that assesses variance in treatment options between an LTC and NTC. The authors address many areas of confounding factors and potential bias in demonstrating the distinct patterns in management between the two types of centers. One issue that would benefit from further input is the decision to be referred to an LTC. The authors point out that some patients are referred to an LTC because it is the closest center and others are referred because their physician thinks they should be considered for transplant. Can the authors break out these two subgroups in the analysis?

Author Response

  1. One issue that would benefit from further input is the decision to be referred to an LTC. The authors point out that some patients are referred to an LTC because it is the closest center and others are referred because their physician thinks they should be considered for transplant. Can the authors break out these two subgroups in the analysis?

We have added the following sentence to paragraph six of our discussion “We are therefore unable to distinguish between LTC patients who were referred specifically for transplant and those who were referred solely for HCC management as the most-suitable HCC referral centre.”

Earlier in the paragraph we discuss at length as follows: “It should be noted also that while some patients have been considered a LTC patient in our study, they may have actually had their initial diagnosis in the community, private health care sector or small peripheral hospital and may have been selected for LTC over NTC referral due to their perceived transplant suitability despite not requiring transplant at time of initial referral. Indeed, between 65% and 75% of patients were referred to centres from outside their direct local hospital catchment, suggesting that selection bias in the preferential referral of transplant−suitable patients to LTC in preference to NTC played a significant role in producing distinct populations with resultant difference in transplant suitability. Explicit data around referral origin, time at managing centre prior to HCC diagnosis, clear documentation regarding perceived transplant suitability and transplant waitlisting was not available for analysis in our study.”